# Elevated Serotonin and NT-proBNP Levels Predict and Detect Carcinoid Heart Disease in a Large Validation Study

**DOI:** 10.3390/cancers14102361

**Published:** 2022-05-10

**Authors:** Sonja Levy, Aoife B. Kilgallen, Catharina M. Korse, Marish I. F. J. Oerlemans, Joost P. G. Sluijter, Linda W. van Laake, Gerlof D. Valk, Margot E. T. Tesselaar

**Affiliations:** 1Department of Medical Oncology, Netherlands Cancer Institute, 1066 CX Amsterdam, The Netherlands; m.tesselaar@nki.nl; 2Regenerative Medicine Centre Utrecht, Circulatory Health Laboratory, University Medical Centre Utrecht, 3584 CX Utrecht, The Netherlands; a.kilgallen@umcutrecht.nl (A.B.K.); j.sluijter@umcutrecht.nl (J.P.G.S.); l.w.vanlaake@umcutrecht.nl (L.W.v.L.); 3Division of Heart and Lungs, Department of Cardiology, Experimental Cardiology Laboratory, University Medical Centre Utrecht, 3584 CX Utrecht, The Netherlands; m.i.f.oerlemans-4@umcutrecht.nl; 4Department of Laboratory Medicine, Netherlands Cancer Institute, 1066 CX Amsterdam, The Netherlands; t.korse@nki.nl; 5Utrecht University, 3584 CS Utrecht, The Netherlands; 6Department of Endocrine Oncology, University Medical Centre Utrecht, 3584 CX Utrecht, The Netherlands; g.d.valk@umcutrecht.nl

**Keywords:** carcinoid heart disease, NT-proBNP, serotonin

## Abstract

**Simple Summary:**

A relevant proportion of patients with neuroendocrine tumors (NET) develop carcinoid heart disease (CHD). This rare cardiac condition leads to worsened survival rates in patients with NET. In this study, we investigated various biomarkers in the blood that could detect and, more specifically, predict which patients are at high risk of developing CHD. In this large study of patients with CHD, we found two biomarkers, NT-proBNP and serotonin, that together are very useful in the prediction and detection of CHD. Moreover, we found cut-off values for NT-proBNP that will aid in the screening of patients with NET, thereby increasing focus on patients at high risk of developing CHD, and releasing patients with a low risk of CHD from burdening screening.

**Abstract:**

Carcinoid heart disease (CHD) is a rare fibrotic cardiac complication of neuroendocrine tumors. Besides known biomarkers N-Terminal pro-B-type natriuretic peptide (NT-proBNP) and serotonin, activin A, connective tissue growth factor (CTGF), and soluble suppression of tumorigenicity 2 (sST2) have been suggested as potential biomarkers for CHD. Here, we validated the predictive/diagnostic value of these biomarkers in a case-control study of 114 patients between 1990 and 2021. Two time-points were analyzed: T_0_: liver metastasis without CHD for all patients. T_1_: confirmed CHD in cases (CHD+, n = 57); confirmed absence of CHD five or more years after liver metastasis in controls (CHD–, n = 57). Thirty-one (54%) and 25 (44%) females were included in CHD+ and CHD– patients, respectively. Median age was 57.9 years for CHD+ and 59.7 for CHD- patients (*p* = 0.290). At T_0_: activin A was similar across both groups (*p* = 0.724); NT-proBNP was higher in CHD+ patients (17 vs. 6 pmol/L, *p* = 0.016), area under the curve (AUC) 0.84, and the most optimal cut-off at 6.5 pmol/L. At T_1_: activin A was higher in CHD+ patients (0.65 vs. 0.38 ng/mL, *p* = 0.045), AUC 0.62, without an optimal cut-off value. NT-pro-BNP was higher in CHD+ patients (63 vs. 11 pmol/L, *p* < 0.001), AUC 0.89, with an optimal cut-off of 27 pmol/L. Serotonin (*p* = 0.345), sST2 (*p* = 0.867) and CTGF (*p* = 0.232) levels were similar across groups. This large validation study identified NT-proBNP as the superior biomarker for CHD. Patients with elevated serotonin levels and NT-proBNP levels between 6.5 and 27 pmol/L, and specifically >27 pmol/L, should be monitored closely for the development of CHD.

## 1. Introduction

Neuroendocrine tumors (NET) are rare, heterogeneous epithelial tumors, with an incidence of 1.09–5.25/100.000 persons per year, occurring primarily in the gastroenteropancreatic tract with the largest group of NET located in the small intestine (SI-NET) [1,2]. In addition, NET can be found in—among others—the lungs and ovaries [3]. Patients with SI-NET often present with regionally advanced or metastatic disease [1,4]. These tumors can secrete vasoactive substances, in particular, 5-hydroxytryptamine (5-HT, also called serotonin) [5]. In some rare occasions, ovarian and bronchopulmonary NET may secrete serotonin [6]. Elevated serotonin can lead to typical symptoms such as flushing, wheezing, and diarrhoea and give rise to the carcinoid syndrome (CS), which occurs in 30–40% of patients with a SI-NET [5,7].

Serotonin is normally metabolized in the liver to the inactive 5-hydroxyindoleacteic acid (5-HIAA); however, the majority of CS patients have liver or retroperitoneal metastases that continuously produce serotonin, which is directly released into circulation [8]. This exposes the heart to high circulating levels of serotonin and causes 20-40% of patients to develop carcinoid heart disease (CHD), as is also shown in a recent cohort of 139 patients with elevated urinary 5-HIAA, where 34.5% developed CHD [7,9,10,11]. CHD is a complication of CS that is characterized by plaque-like deposits, composed of smooth muscle cells and myofibroblasts and an extracellular matrix on the endocardium, leading to fixation and retraction of the heart valves [7,9,10]. Despite advances in therapeutic interventions, CHD is still associated with high mortality rates, especially in patients with advanced valve abnormalities [12,13], even after undergoing valve replacement surgery [8,14].

Currently, as per European guidelines [15], patients with elevated serotonin undergo frequent (1–2 yearly) echocardiography for the detection of CHD, although CHD occurrence is highly variable between patients. Early CHD can be missed or progress to a fulminant form in between screenings, whereas other patients never develop CHD and undergo unnecessary visits to the outpatient clinic. In addition to echocardiographic screening, biomarkers are used to detect CHD. Currently, N-terminal pro B-type natriuretic peptide (NT-proBNP) is the best biomarker in diagnosing and assessing the severity of CHD, with levels of NT-proBNP being significantly higher in CHD patients [12]. NT-proBNP is secreted in response to stretching of the cardiac muscle due to increased pressure and thereby reflects the consequences of CHD, rather than predicting patients at risk for CHD. Serotonin was identified as the key player in the development of CHD, both in human and animal studies [16,17,18]. Yet, besides serotonin, it is assumed that CHD has a multifactorial pathogenesis [7]. Since fibrosis is an important feature of CHD, known mediators of fibrosis were studied in relation to CHD [18,19,20], including activin A, connective tissue growth factor (CTGF), and soluble suppression of tumorigenicity2 (sST2) in several small studies. In these studies, activin A was associated with the presence of CHD with a sensitivity of 87% in a sample of 15 CHD patients [18]; CTGF was shown to be associated with RV dysfunction and valvular regurgitation in 33 patients with NET [19]; and, lastly, sST2 levels that were elevated at CHD diagnosis remained high during and after valve surgery, and only reduced after abdominal surgery for the primary NET [20].

Here, we present the largest cohort of patients to date with blood samples and CHD to investigate the potential use of circulating activin A, CTGF, and sST2 levels as biomarkers associated with the development or presence of CHD, which is confirmed by echocardiography. Our results will be compared to currently used biomarkers known to be associated with the presence of CHD, namely NT-proBNP and serotonin, eventually to identify the superior (combination of) biomarker(s).

## 2. Materials and Methods

### 2.1. Design

In a retrospective single center case-control study, serum samples of patients with CHD (cases) were compared to patients without CHD (controls) to find a classifier to predict and/or detect CHD.

### 2.2. Sample Size

The primary endpoint for samples’ size calculation was based on previous literature and is the sensitivity of the classifier [18]. A power calculation was performed assuming an exact binomial distribution. It was calculated that if the true sensitivity of the classifier is 90%, then a sample of 30 condition positive patients (i.e., CHD patients) will be sufficient to reject the null hypothesis that the sensitivity is 65%, in favor of the alternative that it is higher, with 80% power at a significance level alpha of 0.05 (two-sided).

### 2.3. Patient Selection

The institutional biobank and neuroendocrine neoplasia (NEN) database of the Netherlands Cancer Institute (NKI) stores patient material and clinical data, respectively, of consecutive patients referred to the NKI from 1990 (biobank) and from 2000 (NEN database) until 2021. From these resources, patients with available serum samples and accompanying clinical data were selected. This study was approved by the Institutional Review Board (IRB) under reference IRBm19-137. CHD (CHD+) was defined as the presence of CHD, determined by echocardiography. Controls were selected from the institutional population of patients with SI-NET. No CHD (CHD–) was defined as patients with a SI-NET, radiologically or histopathologically, confirmed liver metastases and elevated serotonin, with no signs of tricuspid or pulmonic regurgitation or other CHD-related right-sided fibrosis of the heart, confirmed by echocardiography, after at least 5 years of follow-up from first occurrence of liver metastases.

Serum samples at two time-points were included, time-point T_0_ and T_1_. Patients were included if either or both time-points were available. For CHD+ patients, T_0_ was defined as the presence of liver- or retroperitoneal metastases and elevated serotonin, with the echocardiographically confirmed absence of CHD, after or at the moment of sample collection. T_1_ was defined as the confirmed presence of CHD, before or simultaneous to sample collection.

For CHD– patients, T_0_ was defined as the presence of liver metastases and elevated serotonin, with the proven absence of CHD, after or at the same moment of sample collection. T_1_ was defined with similar criteria as T_0_, with at least 5 years between the occurrence of liver metastases, before or simultaneous to sample collection. For the prediction of CHD, measurements at T_0_ were compared between CHD– and CHD+ patients. For the detection of CHD, the association of included biomarkers with the presence of CHD was investigated by comparing measurements at T_1_ between CHD– and CHD+ patients. Assays for sST2, CTGF, and activin A were initially performed in selected patients with both T_0_ and T_1_ time-points available (see Figure 1). Based on results from these selected patients, biomarkers were selected for further analysis in all patients.

### 2.4. Echocardiography

Echocardiography reports were reviewed retrospectively, and information extracted to assess the presence of CHD. CHD was defined as at least moderate-to-severe tricuspid and/or pulmonic regurgitation or moderate tricuspid regurgitation identified by the screening cardiologist as related to the NET. Information from the reports was also recalculated to a CHD score by the standardized report recently defined by the European Neuroendocrine Tumour Society (ENETS) CHD Taskforce [21]. Echocardiography was a transthoracic echocardiography (TTE), performed by an experienced cardiologist as per clinical guidelines and standard operating procedure (SOP) for TTE in the Netherlands [22].

### 2.5. Blood Sampling

Peripheral blood from all patients selected for analysis was collected in serum separation tubes, BD Medical, SST, BD Vacutainer. Blood samples were spun down at 1700 g for 10 min to recover the serum. All samples were stored at −80 °C until further analysis.

### 2.6. Enzyme Immunoassays

Serum levels of sST2 and CTGF were analyzed with enzyme-linked immunosorbent assay (ELISA) using the human ST2/IL-33R DuoSet ELISA by R&D Systems (Cat. No: DY523B-05; Minneapolis, MN, USA) and CTGF/CCN2 DuoSet by R&D Systems (Cat. No: DY9190-05; Minneapolis, MN, USA), respectively, according to the manufacturer’s instructions. Activin A serum levels were assessed with an ELISA from RayBiotech (Cat. No: ELH-ActivinA-5, Norcross, GA, USA). sST2, CTGF, and activin A were expressed in ng/mL. Serum serotonin levels were determined by a liquid chromatography tandem mass spectrometry (LC-MS/MS) based assay [23]. Platelet (plt) counts were determined routinely for clinical practice simultaneous to serotonin measurement, and serotonin was expressed as nmol/10^9^plt. Serum levels of NT-proBNP were determined in serum by an electrochemiluminescence immunoassay used on the Modular Analytics E170 (Roche Diagnostics, Mannheim, Germany) and expressed in pmol/L [24].

### 2.7. Statistical Analysis

Descriptive statistics were used for baseline characteristics. Median and interquartile range (IQR) were used for continuous variables; frequencies and percentages were calculated for categorical variables. Fisher’s exact test and the Wilcoxon signed rank test were used for paired comparison within groups and the Wilcoxon rank sum test for comparison between groups. Prior to presentation of the data, logarithmic transformation of sST2, CTGF, and activin A serum samples was performed. The values were derived from linear regression analysis of the standard curve. For the analysis of NT-proBNP and serotonin, non-transformed values were used. Area under the receiver operator curve (AUC) was calculated with a 95% confidence interval (CI). Sensitivity and specificity were calculated for the relevant biomarkers. The case-control design prevented us from calculating the positive and negative predictive values (PPV and NPV, respectively). Statistical analysis was performed using SPSS 26.0.0.1 software (SPSS Inc., Chicago, IL, USA), GraphPad Prism software version 8.3.0 for Windows (GraphPad Software, San Diego, CA, USA) and R statistical software version 4.1.1. The *p*-values were two-sided and considered statistically significant when *p* < 0.05.

Disease specific survival (DSS) was defined as the time from initial diagnosis until NET-related death. Since all patients had stage IV disease at inclusion, patients that died of unknown causes were considered to have died of disease. Patients who were lost to follow-up or alive at end of follow-up were censored. Kaplan-Meier curves were used for analysis of survival. Since inclusion criteria for controls could possibly bias the comparison of survival between CHD and no CHD patients, survival analysis was performed in all consecutive patients with stage IV disease SI-NET referred to the NKI between 2000 and 2019.

## 3. Results

### 3.1. Patients

A total of 114 patients were included, of whom 57 were CHD+ and 57 CHD– patients. No global differences between the CHD and non-CHD group could be found, except standard cardiac medication use in the CHD+ group (49% vs. 28%, *p* = 0.034). Baseline characteristics and a comparison between groups are depicted in Table 1. Median time from NET diagnosis to CHD development was 13 months, ranging from 0 to 142 months. Forty-seven (82.5%) patients underwent annual echocardiographic examination, nine (15.8%) patients bi-annually, and one (1.8%) patient only had the first echocardiography three years after diagnosis of liver metastasis. Six (11%) patients developed CHD after ≥5 years. Tricuspid regurgitation was present in all CHD+ patients, being mild in one (2%) patient, moderate in 13 (23%), and severe in 43 (75%) patients. Pulmonic regurgitation was absent in 4 (7%), mild in 9 (16%), moderate in 11 (19%), severe in 14 (25%), and missing in 19 (33%) patients. Right ventricle dilation was assessed in 48 (84%) patients: cardiac dilation was absent in 11 (19%), mild in 8 (14%), moderate in 18 (32%), and severe in 11 (19%) cases. Echocardiographic characteristics can be found in Table 2. The median CHD score for all CHD patients was 10 (range 3–21), yet individual characteristics were often missing and could not be reported. An overview of CHD score per patient can be found in Appendix A. Seven (12%) CHD+ patients and 13 (23%) CHD- patients had serum samples at two time-points. All patients had a sample at T_1_. A flow diagram of all included patients and the time-points can be found in Figure 1.

### 3.2. Biomarkers in the Prediction of Carcinoid Heart Disease

To predict the development of CHD, measurements at T_0_ were compared between CHD+ and CHD- patients. The T_0_ samples were taken a median of 1 (range 0–7) month after diagnosis of liver metastasis, and a median of 2 (0–6) months prior to echocardiographic absence of CHD for both CHD+ and CHD–. Serotonin levels were equally high in both CHD+ (35.3 nmol/10^9^plt (range 6.77–57.2)) and CHD– patients (29.3 nmol/10^9^plt (range 8.79–49.54)) at (*p* = 0.488) (Figure 2B). Median serum NT-proBNP levels were higher in CHD+ patients (17 pmol/L (range 7–155)) compared to CHD– patients (6 pmol/L (2–23)) (*p* = 0.016) (Figure 2C). Moreover, the AUC for NT-proBNP was 0.84 (95% CI 0.63–1.0) with the most optimal cut-off for NT-proBNP being 6.5 pmol/L, with a sensitivity of 100% and a specificity of 71.4%. Median serum activin A levels in CHD+ patients (0.66 ng/mL (range 0.06–3.75)) and CHD– (0.61 ng/mL [range 0.06–4.93]) were not significantly different (*p* = 0.724) (Figure 2A). Median serum sST2 levels (*p* = 0.867) and CTGF levels (*p* = 0.232) in CHD+ and CHD– patients were not significantly different (Appendix A).

### 3.3. Biomarkers in the Detection of Carcinoid Heart Disease

For detection of CHD, measurements at T_1_ were compared between CHD+ and CHD– patients. For CHD+ patients, T_1_ samples were collected a median of 2 (0–4) months after echocardiographic evidence of CHD. For CHD– patients, T_1_ samples were a median of 2 (range 0–9) months prior to echocardiographic confirmation of absence of CHD, but with a minimum of five years between first diagnosis of liver metastasis and the sample date. Serotonin levels were equally high in CHD+ patients (31.4 nmol/10E^9^plt (range 4.79–93.1)) and CHD– patients (26.7 nmol/10^9^plt (range 7.73–71.9)) (*p* = 0.345) (Figure 2E). Median serum NT-proBNP levels were higher in CHD+ patients (63 pmol/L (range 4–1686)) compared to CHD– patients (11 pmol/L (range 1–213)) (*p* < 0.001) (Figure 2F). The AUC for NT-proBNP was 0.886 (95% CI 0.82–0.96) (Figure 3B). By using the current upper limit of normal (ULN) of NT-proBNP for the absence of cardiac conditions of 35 pmol/L [25,26], a sensitivity for detecting CHD of 77.1% and a specificity of 89.5% would be achieved. In our cohort, a cut-off of 27 pmol/L would provide the optimal threshold for CHD, with a sensitivity of 87.5% and a specificity of 87.7%. Median serum activin A levels between CHD+ (0.65 ng/mL (range 0.04–12.07)) and CHD+ patients (0.38 ng/mL (range 0.06–14.12)) (*p* = 0.0451) were significantly different (Figure 2D). The AUC for activin A was 0.616 (95% CI 0.51–0.72) (Figure 3A), and did not provide an optimal cut-off value for detection of CHD. Median serum sST2 (*p* = 0.694) and CTGF (*p* = 0.955) levels in CHD+ and CHD– patients were not significantly different (Appendix A).

### 3.4. Follow Up and Survival

The median follow-up time for all 114 patients was 7.3 years (IQR 4.3–36.7). Twenty (35%) patients underwent valve replacement surgery. During follow up, 57 (50%) patients died of their NET, and another nine patients (8%) died of unknown causes. In the CHD+ group, 40 (70%) patients died of NET-related causes, and eight (14%) patients died of unknown causes. The cause of death in 11 (28%) CHD+ patients was directly attributable to CHD. Among CHD– known causes. The median DSS in CHD+ patients reached 6.4 years (CI 4.2–8.5); this was 13.7 years (CI 11.7-15.6) in CHD– patients (*p* < 0.001) (Figure 4). Similar results were found when including all consecutive patients with stage IV SI-NET as a control group. A total of 330 patients with stage IV SI-NET were included, with a medium DSS of 14.0 years (CI 8.0–20.0, *p* < 0.001).

## 4. Discussion

In the present study, we aimed to validate if previously investigated circulating biomarkers could detect or predict carcinoid heart disease in the largest cohort of CHD+ patients with blood samples to date. We observed that sST2, CTGF, and activin A did not show a superior association with CHD over currently used biomarkers. Moreover, NT-proBNP levels of 6.5 and 27 pmol/L showed high accuracy for the prediction and detection of CHD, respectively. Furthermore, survival in patients with CHD remains worse in comparison to patients without CHD.

Regarding prediction of CHD, we are the first to identify NT-proBNP to be significantly higher in CHD+ patients, even before the onset of CHD, and can predict the development of CHD. These results suggest that mild to moderate strain on cardiomyocytes might release NT-proBNP before echocardiographic evidence of fibrosis of the right-sided heart can be identified. It is important to note that NT-proBNP is not a marker that shows the causal molecular pathway of the pathogenesis of CHD, and is therefore rather a sensible early diagnostic marker than a true predictor. Nevertheless, since NT-proBNP is elevated in patients that will develop echocardiographic CHD in the future, it has the capability to differentiate at baseline between patients who are at risk of developing CHD and those that are not. Because of these strong predictive abilities, we have chosen to call it a predictor. 

Regarding detection of CHD, NT-proBNP expression is significantly elevated in CHD+ patients and directly associates with CHD severity [24,27,28,29]. For instance, in a cohort of 187 patients with NET and liver metastases, of whom 37 had CHD, NT-proBNP was found to be to have an AUC of 0.82 [28]. Our results confirm that NT-proBNP outperforms other biomarkers for CHD, and further identify that a cut-off of 27 pmol/L has the best accuracy of detecting CHD. With these findings, we argue that clinicians could make a more accurate distinction of patients that would benefit from (more frequent) echocardiographic screening, and which would not. For instance, patients with NT-proBNP levels >6.5 pmol/L could undergo echocardiography 1–2 yearly as per current guidelines, whereas patients with NT-proBNP levels <6.5 pmol/L could be released from echocardiographic screening, but be followed only with active monitoring of NT-proBNP levels. Moreover, patients without echocardiographic signs of CHD, but with NT-proBNP levels >27 pmol/L, could possibly benefit from more active screening than is currently advised by European guidelines [15], for instance, by six monthly echocardiography.

Activin A levels differed significantly between CHD+ and CHD– patients at T_1_. Despite this, we found that activin A was not able to provide an optimal cut-off level for CHD in our cohort. In the study by Bergestuen, et al., activin A levels ≥0.34 ng/mL were found to be associated with an increased risk of developing CHD in 15 patients [18]. The positive results for activin A in that study may have been caused by the small number of patients included. Most CHD patients included in this study cohort had moderate to severe or severe regurgitation and thickening of the tricuspid valves (TV) or pulmonary valves. It is hypothesized that activin A may reach a threshold value to initiate the molecular pathways associated with fibrosis, and not play a role in disease progression [18]. This may explain why, although elevated in CHD+ patients, we were not able to identify a cut-off value for detection of CHD, since this threshold may have been reached in moderately elevated levels of activin A. Nevertheless, it remains unknown why some patients would develop CHD above this threshold, and others do not.

Serotonin is still regarded as the best clinical tool in identifying patients at risk of CHD. However, it can be limited in providing optimal accuracy in diagnosing CHD since not all patients with elevated serotonin develop CHD. We did not find an association between higher serotonin levels and CHD, as was identified previously [28,30,31]. A recent review concluded that elevated 5-HIAA levels were associated with CHD and with higher mortality [32]. Yet, previous studies mostly compared CHD patients with NET patients, with or without elevated serotonin, whereas we refined our inclusion criteria and specifically selected controls with confirmed liver metastases and elevated serotonin. Consequently, this selection provided a more homogeneous group of patients to study, but it prevented us from comparing serotonin levels to patients with a NET in general. Nevertheless, the fact that we found equal groups of patients with and without CHD during the inclusion period, this again suggests that elevated serotonin may not be the only factor that contributes to CHD, but an unknown causal factor is involved in the development of CHD. This implies that the management of CHD patients should not only be aimed at reducing serotonin levels by known methods such as somatostatin analogues or debulking surgery, but also at early detection and intervention for CHD.

We found that patients with CHD had a worse survival compared to patients without CHD. This was also confirmed by other studies investigating the prognosis of patients with CHD [32,33,34]. Indeed, the percentage of deaths directly attributable to CHD (27.5%) seems to make up the difference in survival between patients with and without CHD. This stresses the urge for early recognition and possible intervention for CHD in patients with elevated serotonin.

There are several limitations worth mentioning. Firstly, we used a cut-off value of five years as a criterion for the selection of controls. It is possible that patients in the CHD– group could yet develop CHD during follow-up. Nevertheless, in our CHD+ cohort, nearly 90% of patients developed CHD within five years of liver or retroperitoneal metastases. Therefore, it is unlikely that the number of CHD– patients who could possibly still develop CHD would be large enough to bias our results. Secondly, our sample size calculations were based on the detection of CHD, and not on prediction of CHD. Moreover, a total of 31 patients had samples at T_0_, which might be insufficient to identify the optimal cut-off level of NT-proBNP adequately for the prediction of CHD. Nonetheless, our results are the first to indicate a cut-off value for the detection of CHD, and provide evidence that NT-proBNP levels are significantly higher in patients with CHD, even before any abnormalities can be found by echocardiography. These results stress the need for adequate monitoring of patients with elevated serotonin, even with moderately elevated NT-proBNP levels.

A major strength of this study is the large sample size. This is the largest study to date to investigate patients with CHD and possible associated biomarkers. Moreover, we were able nearly to double the sample size initially calculated for this study, therefore increasing the statistical power.

## 5. Conclusions

In conclusion, in this largest validation study of biomarkers for CHD to date, we found that sST2, CTGF, and activin A are not useful in predicting or detecting CHD over currently used biomarkers. NT-proBNP, in the presence of elevated serotonin, remains the best suited biomarker in clinical practice. This is the first study that provides structured guidance in the management of patients with serotonin producing NET. Patients with NT-proBNP values below 6.5 pmol/L could likely be released from echocardiographic screening, whereas patients with NT-proBNP values above 6.5 pmol/L could undergo screening as per current guidelines. Moreover, patients with NT-proBNP above 27 pmol/L should be monitored even more closely for the development of CHD. Patients with CHD have a worse disease specific survival compared to patients without CHD. Future studies should focus on elucidating the molecular mechanisms of the development of CHD and further identify patients at risk thereof.

## Figures and Tables

**Figure 1 cancers-14-02361-f001:**
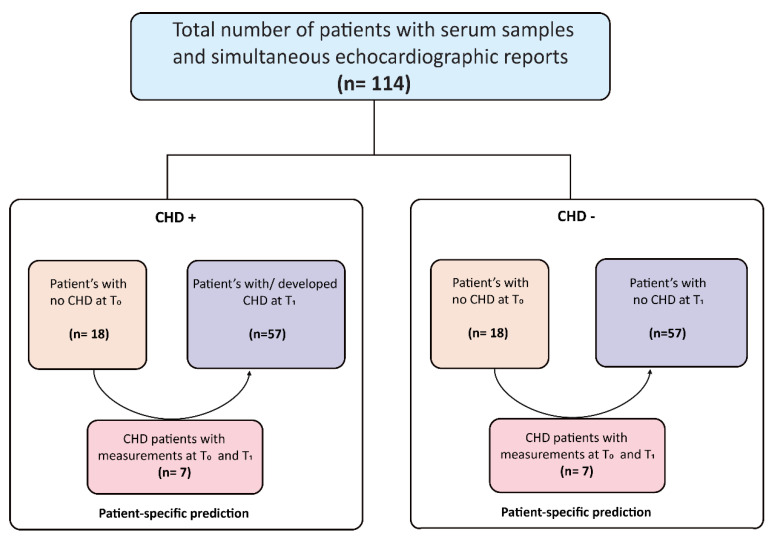
Flow diagram of the included study patients in the CHD+ or CHD- group at T_0_ and T_1_.

**Figure 2 cancers-14-02361-f002:**
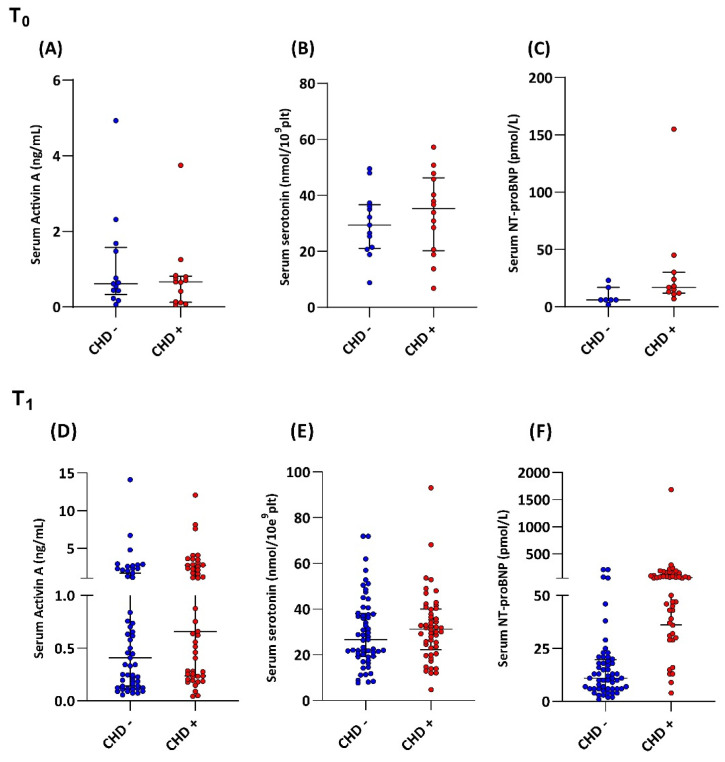
Serum Activin A (**A**), serotonin (**B**), and NT- pro BNP (**C**) levels in CHD+ patients and CHD- patients in the prediction (T_0_) and detection of CHD (T_1_) (**D**–**F**).

**Figure 3 cancers-14-02361-f003:**
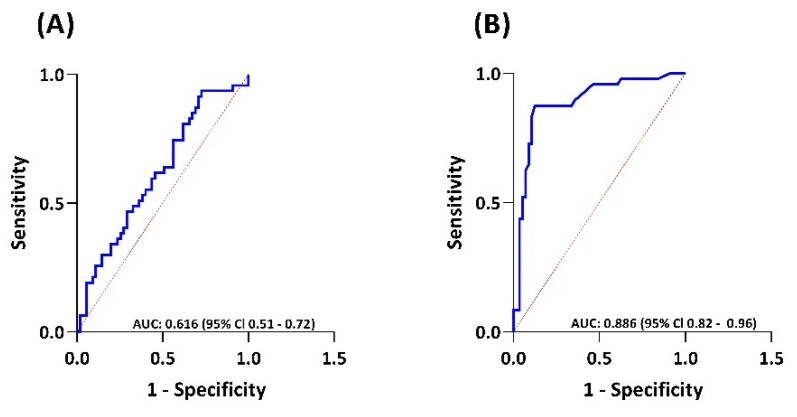
(**A**) Receiver operator characteristic (ROC) curve of activin A in the detection of CHD at timepoint T_1_, between CHD+ and CHD- patients (AUC 0.616 (95% CI 0.51–0.72), *p* = 0.0451); (**B**) ROC-curve representing the ability of NT-pro BNP to detect CHD between CHD+ and CHD- patients at T_1_ (AUC 0.886 (95% CI 0.82–0.96), *p* < 0.001). AUC: area under the curve. CI: confidence interval.

**Figure 4 cancers-14-02361-f004:**
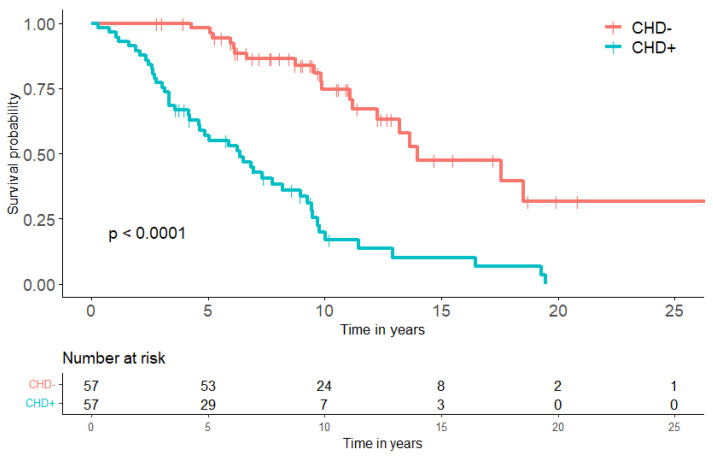
Kaplan–Meier curves for disease specific survival. Logrank test was performed for comparison between groups. CHD: carcinoid heart disease.

**Table 1 cancers-14-02361-t001:** Baseline characteristics of all included patients.

	Group		
Characteristic	CHD− (n = 57)	CHD+ (n = 57)	*p*-Value
Sex, n (%)			
Male	26 (45.6)	32 (56.1)	
Female	31 (54.4)	25 (43.9)	0.349
Median age at diagnosis, years (range)	57.9 (32.3–76.9)	59.7 (26.8–81.7)	0.290
Primary tumor, n (%)			n/a †
Small intestine	56 (98.2)	37 (64.9)	
Ovarium	0	2 (3.5)	
Lung	0	2 (3.5)	
Unknown	1 (1.8)	16 (28.1)	
Patients receiving treatments, n (%)	16 (28.1)	28 (49.1)	0.034 ¥
Beta blockers	8 (14.0)	8 (14.0)	
ACE-inhibitor	2 (3.5)	5 (8.8)	
Calcium antagonist	7 (12.3)	5 (8.8)	
Nitrates	0	2 (3.5)	
ARB	0	5 (8.8)	
Diuretics	1 (1.8)	21 (36.8)	
Median CHD score (range)	n/a	10 (3–21)	n/a

*p*-values show Fisher’s exact test for comparison between the patient groups. Medication prescribed to patients included in the study at any moment during follow up. ACE: angiotensin converting enzyme, ARB: angiotensin receptor blocker. † Comparison irrelevant since controls were selected from a cohort of patients with small intestinal net. ¥ For comparison of cardiac medication yes/no between groups.

**Table 2 cancers-14-02361-t002:** Echocardiographic characteristics of all patients with confirmed carcinoid heart disease (CHD).

Echocardiographic Characteristic	CHD Patients (n = 57)
TV regurgitation, n (%)	
Mild	1 (1.8)
Moderate	12 (21.1)
Severe	44 (77.2)
TV leaflet thickening, n (%)	
None	5 (8.8)
Mild	10 (17.5)
Moderate	22 (38.6)
Severe	37 (64.9)
Missing	20 (35.1)
PV regurgitation, n (%)	
None	4 (7.0)
Mild	8 (14.0)
Moderate	11 (19.3)
Severe	13 (22.8)
Missing	21 (36.8)
RV dilation, n (%)	
None	10 (17.5)
Mild	7 (12.3)
Moderate	17 (29.8)
Severe	11 (19.3)
Missing	12 (21.1)
RV impairment, n (%)	
None	32 (56.1)
Mild	5 (8.8)
Moderate	2 (3.5)
Severe	1 (1.8)
Missing	17 (29.8)
MV regurgitation, n (%)	
None	6 (10.5)
Mild	22 (38.6)
Moderate	7 (12.3)
Severe	3 (5.3)
Missing	19 (33.3)
AV regurgitation, n (%)	
None	13 (22.8)
Mild	15 (26.3)
Moderate	3 (5.3)
Severe	0
Missing	26 (45.6)

TV: tricuspid valve, PV: pulmonic valve, RV: right ventricle, MV: mitral valve, AV: aortic valve.

## Data Availability

The data presented in this study are available on request from the corresponding author.

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
