# Peer review of "Elevated Serotonin and NT-proBNP Levels Predict and Detect Carcinoid Heart Disease in a Large Validation Study"

_cancers, 2022, doi:10.3390/cancers14102361_

Round 1

Reviewer 1 Report

The authors have undertaken an important (and difficult) study on some biomarkers of CHD. I have some questions to them that I list below

The main one is that I dont agree you can call proBNP a predictor by definition; it is a marker of cardiac muscle overload (damage already installed). You could say it is sensible diagnostic marker but not a predictor because it is not there at baseline before CHD onset.

Other comments:

1-Controls were selected from the institutional 112 population of patients with SI-NET: why not other primary NET? 

2- please describe the echo intervals. anually? this may influence CHD onset

3- Describe time windows of sample collection , echocardiograms, imaging to define metastatic disease (one month a part? More?)

4- How did you analyze pts who died before 5y of FUP ? were these excluded from the study if they had not develop CHD? if so, this may have introduced a potentially relevant bias. please comment and inform the N of excluded pts in this situation.

5-There is one multicenter study on prognostic factors of CHD, with 139 pts. the authors found that elevated 5HT, liver involvement and non SI NET to be associated with CHD. Why did you decide to omit it? I think it would enhance your discussion. 

Suggestion:

Have you thought of looking at marker of early onset of CHD? E.g. 1-2 years from T0

Reviewer 2 Report

The authors suggest that elevated serotonin and NT-proBNP levels predict and detect carcinoid heart disease. The investigation is so interesting, however, I have some concerns.

1.You claim that the onset of CHD should be closely monitored, but what exactly should be done?

  1. How does the use of circulating Activin A, CTGF and sST2 levels correlate with NT-proBNP and serotonin?
  2. What is the novelty of the current study?
  3. Why were sST2, CTGF, and Activin A selected?
  4. Not useful for acute myocardial infarction?

Round 2

Reviewer 2 Report

The authors show that serotonin and NT-proBNP levels predict and detect carcinoid heart disease. The study is so interesting, however, I have somce concerns to be discussed.

1. What is the best way to keep serotonin levels and NT-proBNP levels below 27 pmol/L?
2. Please highlight any new findings from this study.
3. What kind of experiments in vitro would prove the clinical case in this study?
4. Are serotonin and NT-proBNP levels involved in liver metastasis?

Author Response

Please see the attahment.

Round 3

Reviewer 2 Report

The authors answered well to my concerns, so the manuscript is suitable for publication.